# Combining Cortical Voltage Imaging and Hippocampal Electrophysiology for Investigating Global, Multi-Timescale Activity Interactions in the Brain

**DOI:** 10.3390/ijms23126814

**Published:** 2022-06-19

**Authors:** Rafael Pedrosa, Chenchen Song, Thomas Knöpfel, Francesco Battaglia

**Affiliations:** 1Donders Institute for Brain Cognition and Behaviour, Radboud University, 6525AJ Nijmegen, The Netherlands; fpbattaglia@gmail.com; 2Laboratory for Neuronal Circuit Dynamics, Imperial College London, London W12 0NN, UK; song@knopfel-lab.net

**Keywords:** cortical activity, hippocampus, voltage imaging, electrophysiology, mesoscale, GEVI

## Abstract

A new generation of optogenetic tools for analyzing neural activity has been contributing to the elucidation of classical open questions in neuroscience. Specifically, voltage imaging technologies using enhanced genetically encoded voltage indicators have been increasingly used to observe the dynamics of large circuits at the mesoscale. Here, we describe how to combine cortical wide-field voltage imaging with hippocampal electrophysiology in awake, behaving mice. Furthermore, we highlight how this method can be useful for different possible investigations, using the characterization of hippocampal–neocortical interactions as a case study.

## 1. Introduction

Hippocampal–cortical interactions have been widely studied with the aim of understanding their role in supporting memory [1,2,3,4]. Yet, most studies have focused on the interplay between the hippocampus and selected cortical areas with major anatomical connections to the hippocampus, such as the entorhinal or the medial prefrontal cortex [4,5,6,7]. Less is known about the interrelation between hippocampal activity and activity phenomena spanning large portions of cortical space [8,9,10]. Cortex-wide activity is classically studied with imaging techniques that rely on indirect correlates of neural activity, such as the blood-oxygen-level-dependent (BOLD) or intracellular calcium signals, whereas most of our knowledge of the hippocampus comes from microelectrode recordings [6,11]. While classical imaging methods allow for excellent spatial coverage, they suffer from limited spatial resolution. Microelectrode techniques, on the other hand, provide recordings with excellent temporal resolution but limited spatial information [12]. In particular, conventional widefield imaging using genetically encoded calcium indicators (GECIs) presents a significant temporal discrepancy in activity when compared with electrophysiology, due to the distinct nature of the recorded signals [13,14]. Cortical widefield voltage imaging using genetically encoded voltage indicators (GEVIs) in head-fixed mice provides a large-scale view on cortical activity with both high spatial and high temporal resolution, and it can be combined with electrode-based electrophysiology [15]. Here we describe the combination of cortical voltage imaging with hippocampal electrode recordings to simultaneously monitor activity both from the cortex and the hippocampus during rest and traditional behavioral tasks. Specifically, we give examples of how this combination can contribute to questions focused on sensory-hippocampal integration and hippocampal–cortical interaction.

## 2. Results

### 2.1. Simultaneous Cortical Voltage Imaging with Hippocampal Electrophysiology 

To investigate the interactions between the hippocampus and the cortex, here we used cortex-wide voltage imaging in combination with electrophysiological recording using a silicon probe in the hippocampal CA1 (Figure 1) [2,3,16]. We used transgenic mice that express the transgene for chimeric VSFP Butterfly Y/R (chiVSFP Butterfly) under the tetO promoter. Crossing with CaMK2A-tTA mice resulted in chiVSFP Butterfly expression in neocortical and hippocampal pyramidal neurons [16]. By replacing a segment of the Ciona intestinalis voltage sensing domain by a homologous portion of the fast activating and deactivating Kv3.1 potassium channel, chiVSFP Butterfly was developed from earlier VSFPs [16,17,18]. This modification accelerated the GEVI response dynamics, with chiVSFP Butterfly readily following membrane voltage oscillations at frequencies up to at least 200 Hz. In contrast to monochromatic GEVIs, chiVSFP Butterfly reports membrane voltage changes in two fluorescence bands with opposite fluorescence changes. This allows for ratiometric measurements that facilitate correction for hemodynamic and pH-related confounds inherent to monochromatic (single-wavelength) GEVI imaging [19]. 

First, we recorded two vital states, in vivo (in resting state) and ex vivo (following terminal euthanasia) from the same mouse to compare the signal-to-noise ratio. As expected, in the ex vivo condition, the neocortex did not show any activity, whereas in the living brain spontaneous fluctuations were observed in the neocortex (Figure 2A). As was also expected, the probabilistic distribution of cortical transients in the in vivo state adheres to the power law, whereas that of the ex vivo state is reflective of noise. Similarly, in the in vivo state, in the hippocampal electrophysiological recordings that were synchronized to cortical voltage imaging, spontaneous sharp-wave ripple (SWR) events can be observed in the hippocampal pyramidal layer which accompanies the cortical transients (Figure 2A). In the ex vivo state, there is an absence of hippocampal activity, as expected. 

The impact of movements in the active behavioral state causes a strong modulation in the cortex-wide neural activity [20,21,22,23,24]. In the rodent hippocampus, the theta frequency band (5–10 Hz) is a hallmark of active states such as walking, running and sniffing. In line with previous studies [22,24], we found that cortical activity in multiple brain areas increased during bouts of treadmill running. Interestingly, in coexistence with hippocampal theta, cortical activity, mostly in the primary motor cortex (M1), seems to be more tightly related to running (Figure 2C). When correlating hippocampal theta power with the cortical activity, we also found M1 activity preceding episodes of increased hippocampal theta oscillation (Figure 2D).

### 2.2. Layer-Specific Hippocampal Signal during Integrative Visual Information Process

In rodents, eye movements have also been linked to spatial navigation processes [25,26,27,28]. Visual information is crucial for spatial navigation, where instantaneous visual input constantly updates egocentric and allocentric spatial information, and gaze direction is a vital input to that computation. Here, we show that stimulus-evoked voltage activity spreads from the visual cortex to higher cortical areas such as the retrosplenial, cingulate and parietal cortices (Figure 3A,B). Additionally, we show that combining eye-tracking with hippocampal probe recordings allows the investigation of hippocampal layer-specific activity associated with eye saccade movements (Figure 3C). Interestingly, we specifically observed an emergence of gamma-band activity in the stratum lacunosum moleculare following the saccade movement. Additionally, this modulation seems to be different across layers, which suggests that the saccade movement of the eye associates with local circuits within the hippocampus. Finally, when correlating gamma-band activity in the stratum lacunosum moleculare with the imaged cortical activity, we observed a correlation between a set of medial to lateral areas (parietal, primary and secondary lateral visual cortices; primary auditory cortex) that precedes medium gamma oscillation in the hippocampal CA1 (Figure 3D). During episodes of medium gamma oscillations, this correlation seems to be centered on more medial areas (primary and secondary motor, sensory and retrosplenial cortices), which may be related to the locomotion of the animal.

### 2.3. Hippocampal Neuronal Interrelation with Cortical Modules

The hippocampus has several efferent and afferent projections connecting with different brain regions including the cortex, and hippocampal-cortical communication is crucial for processes like archiving memory traces from short-term cortical representations into long-term hippocampal storage [4,29]. Combining cortical voltage imaging with hippocampal silicon probe recordings thus provides a unique opportunity for studying this connection in vivo. Here, we used 16-channel silicon probes to detect hippocampal CA1 activity (Figure 4A) in awake, head-fixed animals that have been implanted with a transcranial window for cortical voltage imaging. We detected multiple neurons with distinct waveforms (Figure 4B). Next, we computed spike-triggered averages of the neocortical activity associated with hippocampal neuron spiking activity, and observed different cortical maps associated with the firing activity from different neurons from the CA1 pyramidal layer (Figure 4C). Here, we demonstrate that this combined approach offers the potential to examine hippocampal-cortical associations in detail.

### 2.4. Cortical Microstimulation and Hippocampal Electrophysiology

Next, we demonstrate the technical feasibility of observing both cortical voltage activity and hippocampal electrophysiological activity in response to cortical microstimulation in vivo. To this end, we electrically stimulated the retrosplenial (RS) and somato-sensory (SS) cortices individually with varying stimulation intensities (20, 50, 100 and 200 µA), and monitored how these two cortical structures interact with both the rest of the cortical hemisphere and with hippocampal oscillations at different frequencies in a resting-state mouse (Figure 5A).

Following RS or SS stimulations at 200 µA, we observed a stimulated area displaying an instantaneous burst of activity followed by a spreading hyperpolarization lasting for about one second in the cortical imaging data (Figure 5B). Looking at stimulation effects on the hippocampal spectral activity at different intensities (20, 50, 100 and 200 µA), we found that RS stimulation (suppression) caused an interference on the hippocampal LFP (mostly at frequencies between 5 and 40 Hz) (Figure 5C). This hippocampal oscillatory interference is proportional to the size of the current (and consequently RS suppression) induced in the RS area (Figure 5D). Conversely, SS stimulation did not show significant differences in the hippocampal LFP (repeated-measures ANOVA; N = 50 stimuli for each condition). Combining cortical imaging, cortical microstimulations and hippocampal LFP, our observations suggest that a causal relationship may exist between the retrosplenial cortex and hippocampal CA1, but further investigation is needed to establish if this is due to cortical inputs into the hippocampus, or due to antidromic activation and the suppression of hippocampal afferents in the retrosplenial cortex.

## 3. Discussion

A common caveat of many system-level studies is the assumption that specific cognitive responses only involve specific brain areas. This is because in vivo recordings are commonly designed to target only a few areas, leaving the contributions of the rest of the brain aside. Here, we advocate that widefield voltage imaging can be a valuable approach for observing multiple cortical areas simultaneously in awake, behaving animals. Compared to indicators of other brain activity proxies, such as calcium and glutamate, voltage activity has a faster temporal resolution, similar to that of electrophysiology, which allows causal analysis in combination with electrophysiological data. 

It is important to emphasize that the combination of electrophysiology with cortical voltage imaging is flexible and may be adapted to different scientific questions. Here, we demonstrated the feasibility of combining cortical voltage imaging with hippocampal electrophysiology to start yielding new insights into the hippocampal –cortical relationship associated with visual processing. 

Visual information plays an important role in spatial navigation [30,31]. Particularly, identifying landmarks and borders are processes that are needed for spatial cognition and require information from primary visual areas to reach higher areas in the brain’s organizational hierarchy such as the hippocampus in the brain, where a cognitive representation of the environment may be generated. Nevertheless, how visual signals modulate spatial representations in the hippocampus and how information propagates to the hippocampus is still unknown. To investigate this, we advocate that widefield cortical voltage recordings may provide important details on the visual integration mechanism [16,32,33]. Towards this goal, we show here that the activity from the visual cortex propagates to other higher cortical areas that are interconnected, such as the hippocampus, retrosplenial, cingulate and parietal cortices [32,34]. Additionally, we observed that saccadic movement relates mostly to medium gamma in the stratum lacunosum moleculare. Correlating this hippocampal gamma activity with cortical activity, we found that activity in a set of cortical areas (among those areas the visual cortex) precedes the medium gamma in the hippocampus. Thus, with our combination of techniques we could demonstrate a novel interaction between eye–hippocampus–cortex in the awake state.

Hierarchically, the hippocampus is seen as an important central area for the memory process. It is composed of several efferent and afferent projections scattered throughout the brain, among them the cortex. The classic hypothesis that the hippocampus retains short-term memories and indexes them to the cortex for long-term memory highlights the importance of hippocampal–cortical communication [4,29]. Prior to this process, a short-term memory emerges from the encoding of sensory information and its adaptation to the hippocampus. In this sense, the spike modulation resulting from the hippocampal–cortical interaction can address this computation. Here, we showed that activity in the hippocampus can have different temporal and spatial dynamics to cortical activity. On a broad neocortical level, the reactivation of memory has previously been observed [33,35]. Together with our result, we can speculate that information about hippocampal events such as replays or the formation of cell assembles are transmitted to specific cortical circuits.

We also demonstrated a combination between cortical voltage activity and hippocampal LFP during microstimulations of the retrosplenial or somatosensory cortex. Our results show that gamma oscillation in the hippocampus is suppressed by the inhibition caused specifically in the retrosplenial cortex by the stimulation, and that this suppression is correlated with the intensity of the current in the retrosplenial cortex. Previously, we reported that retrosplenial cortex activity interacts with gamma oscillation in CA1 during sleep, and that this interaction has a temporal relationship from the cortex to the hippocampus [3]. This, together with our result here, indicates that retrosplenial-CA1 communication has a direct influence on the hippocampal LFP. 

In conclusion, the hippocampal–cortical relationship is recognized in neuroscience for playing a fundamental part in the memory consolidation mechanism. Despite this, the study of basic interactions between these two structures presents functional gaps, which in part is due to technical limitations. In the upcoming years, cortical wide-field voltage imaging may contribute to the understanding of hierarchical neocortical processes in different behavioral states. Combined with hippocampal LFP, this approach has the potential to clarify the fundamental circuitry of cognitive processes such as memory consolidation.

## 4. Materials and Methods

### 4.1. Animals

We used CaMK2A-tTA;tetO-chiVSFP transgenic mice (3 to 6 months and weighing 25–35 g) [16,18]. This line of transgenic mouse expresses a GEVI specifically in pyramidal neurons across all cortical layers. Macroscopic epifluorescence imaging restricts the optical access to the superficial cortex [16,17,32,33]. Animals were housed under a 12 h light/dark cycle with ad libitum water and food access. 

### 4.2. Surgical Procedure for Combining Wide-Field Imaging with Hippocampal Electrophysiology

To record cortical optical voltage imaging in combination with hippocampal electrophysiology (N = 4 mice), we performed a surgical procedure to expose the scalp and chronically implant a 16-channel linear probe (Atlas Neuroengineering, Belgium, 50 μm spacing between recording sites) into the hippocampus. For surgeries, animals were anaesthetized and placed in a stereotaxic frame with a nasal mask delivering isoflurane at 0.5–1.5%. During the entire surgery, the body temperature was maintained by a heat-pad under the body around 37 °C and the breathing rate at 0.5–1 Hz. Then, 2% lidocaine (50 μL) was injected subcutaneously at the incision zone. We successively exposed a wide area of the scalp (Figure 1A), and thinned the skull of the entire right hemisphere with a surgical drill in order to remove the skull capillaries to reduce light scattering. Additionally, two screws were implanted into the skull of the left hemisphere to improve the mechanical stability of the head-plate; a third screw was also implanted in the superficial region of the cerebellum in the same hemisphere to be used as a ground and reference. For hippocampal electrophysiology recording, a 16-channel linear probe (Atlas Neuroengineering, Belgium, 50 μm spacing between recording sites) was chronically implanted into the right hemisphere (ipsilateral of the imaged cortex) (Figure 1B). The probe was initially placed in +2.4 mm ML and −4.3 mm AP at a 60-degree angle, and then lowered and fixed at ~2.2 mm depth. Finally, the head-plate was carefully placed and fixed onto the skull with acrylic cement.

### 4.3. Combining Wide-Field Voltage Imaging with Hippocampal Electrophysiology

We performed synchronized cortical voltage imaging and hippocampal LFP recording from the mice after post-operative recovery. The optical configuration for voltage imaging is as previously described, using a widefield dual-emission macroscope (Figure 1C,D) [16]. Fluorescence excitation was provided by high-power halogen lamps (Brain Vision, Moritex), and we simultaneously measured both mKate2 (GEVI FRET acceptor) and mCitrine (GEVI FRET donor) epifluorescence emissions using two synchronized sCMOS PCO edge 4.2 cameras in combination with Leica PlanAPO1.0 and Leica PlanAPO1.6 lenses. The optics used were: mCitrine excitation 500/24 and emission FF01-542/27, mKate2 emission BLP01-594R-25 with beam splitters 515LP and 580LP (all Semrock). Image series were acquired from the right cortical hemisphere at 50 Hz frame rate with 375 × 213 pixel resolution (~60 pixel/mm) and 12 bit sampling depth, in blocks of 10 min.

To compute the optically measured voltage activity, a gain-equalized ratio (ΔR/R) between the mCitrine–mKate2 fluorescence emission was calculated (R = mCitrine/mKate2), then normalized as ΔR/R = (R(t) − Rmean)/Rmean [36,37]; Rmean was the session-averaged ratio. We isolated and subtracted the heartbeat frequency components in addition to a high pass filter >0.5 Hz to remove residual hemodynamic signals [36,37]. To quantify the cortical voltage fluctuations, we detected the activity transients using a similar approach to that reported by Scott et al. [38], where the global state represented by the active states of each pixel has to cross a certain threshold defined as the mean of the total activated frames. 

Synchronized to the cortical voltage imaging, local field potential (LFP) from the hippocampal CA1 was acquired using a 16-channel headstage (Intan Technologies, RHD2132, Los Angeles, CA, USA) and an Open Ephys system. The raw signal was recorded with 0.1–7500 Hz filtering and sampled at 20 kHz, and then down-sampled to 1 kHz for analysis. We used kilosort 2.0 [39] for spike sorting. Theta power was computed from the envelope of the filtered LFP signal between 5–10 Hz.

### 4.4. Visual Stimulus Presentation

Visual stimuli were presented on a monitor positioned 10 cm from the left eye of the mouse. We presented 50 gratings at 45 degrees during 1 s with random intervals between 1 and 2 s.

### 4.5. Eye-Tracking and Pupil Detection

To detect the saccade movements of the eye of the mouse during recording, we used a fixed camera (Basler aca1920-150 um, objective with f = 20 mm) directed to the face of the head-fixed mouse with an infrared illumination, and acquired images at 20 Hz.

We then used DeepLabCut [40] to label the pupil size detected in each behavioral imaging frame. From the high-movements of the circle centroid in the pupil, we obtained saccade times by the thresholding eye speed from the resultant vector of the movements of the eyes. This method has been previously described [41].

### 4.6. Cortical Microstimulation with Combined Cortical Voltage Imaging and Hippocampal Electrophysiology

For cortical microstimulations, we used a bundle of two tungsten electrodes (50 μm diameter) that were staggered to create a vertical distance of ~250 μm between the tips. The implant surgical procedure is as described above, with the addition of a chronic implantation of two stimulation electrode bundles into the retrosplenial (RS) and somato-sensory (SS) cortex before the head-bar fixation. For RS implantation, we first performed a small craniotomy on the right hemisphere of the skull at coordinates +0.5 mm (ML) and −2.5 mm (AP), and implanted the bundle electrode in +0.5 mm (DV) and at 45 degrees tilt towards the midline (outside of the visual field of the camera for the cortical imaging). Then we performed a small craniotomy on the right hemisphere at coordinates +3 mm (ML) and +1 mm (AP) above the SS cortex. The bundle electrode in this case was also implanted +0.5 mm (DV), but angled at 30 degrees tilt towards the lateral side of the brain.

After 10 days of post-operative recovery, awake animals were placed into the recording setup for cortical stimulation in combination with cortical voltage imaging and hippocampal electrophysiology. We used an Arduino UNO to send 50 square pulses of 10 milliseconds, with inter-stimulus interval randomized between 1.5–2.5 s, controlling a stimulus isolator (World Precision A365), which was also connected to Open Ephys for the synchronization. Stimulation was provided at intensities of 20, 50, 100 and 200 µA.

## Figures and Tables

**Figure 1 ijms-23-06814-f001:**
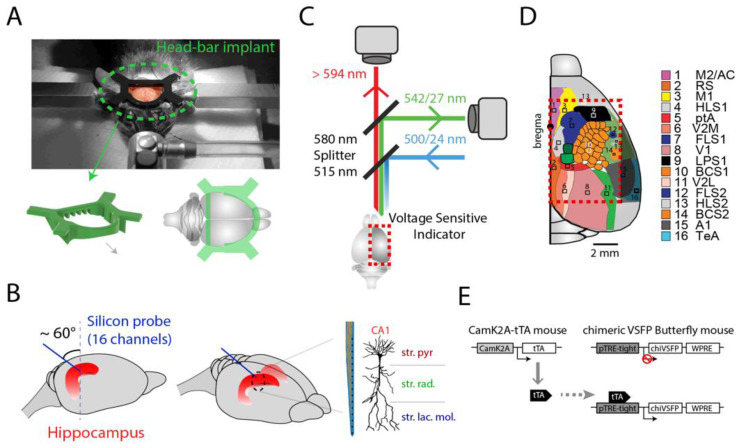
Implant procedures for combined widefield cortical imaging and hippocampal electrophysiology recordings. (**A**) Head-fixation bar implantation on a mouse placed in the stereotaxic apparatus. (**B**) Schema of high-density silicon probe (16 channels) implantation into the hippocampus. With 50 μm spacing between the contacts, the probe can cover all layers in CA1. Note that the implant is in the posterior part of the cortex, specifically close to the lambdoid suture, thus leaving the entire cortical area unobstructed in the optical imaging field of view. (**C**) Dual emission macroscope used for widefield cortical optical voltage imaging. The fluorescence emissions of the voltage indicator from the superficial cortical tissue was recorded by two synchronized cameras, and subsequently used for ratiometric analysis. (**D**) Topographic map of the multiple areas that can be imaged together by the macroscope (topographic image adapted with permission from Ref. [2]). (**E**) Schematic diagram of the GEVI line of transgenic mice used; CaMK2A-promoter-controlled chimeric VSFP Butterfly (chiVSFP) expression in pyramidal cells (CaMK2A-tTA;tetO-chiVSFP) [16].

**Figure 2 ijms-23-06814-f002:**
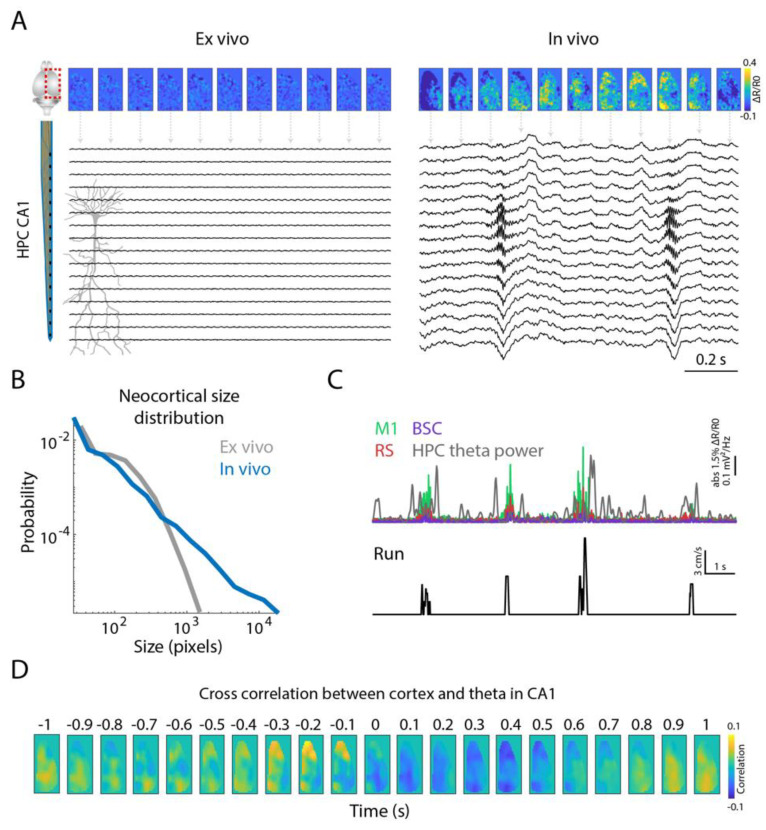
Signal quality. (**A**) Examples of cortical voltage imaging and hippocampal CA1 electrophysiology during ex vivo and in vivo states of an animal. (**B**) Probabilistic distribution of neocortical transients based on the spatial extent of the activity fluctuations (size) for ex vivo and in vivo states. As expected, the transients in the in vivo state obey a distribution that is more likely to follow a power-law scale. Smaller transients, also present in the ex vivo preparation, are likely due to noise. (**C**) Simultaneously recorded absolute ΔR/R cortical activity and hippocampal theta power during active (run periods) and quiet behaviors. Note that during the running periods the animal presented a higher modulation of the cortical voltage signal, which is similar to the increased theta power in the hippocampus. The labels are defined as primary motor cortex (M1), retrosplenial cortex (RS), barrel cortex (BSC) and hippocampal (HPC) theta power. (**D**) Cross correlation between the cortical pixels and theta power in the hippocampal stratum pyramidale.

**Figure 3 ijms-23-06814-f003:**
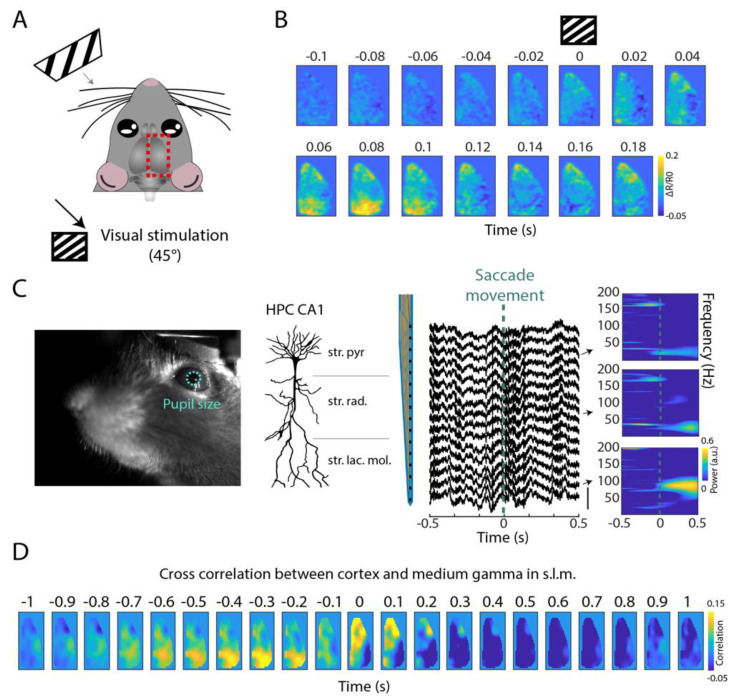
Visual information propagates to cortical association areas and the hippocampus. (**A**) Schematic of the visual stimuli paradigm. (**B**) Average cortical voltage activity profile during specific visual stimulation pattern (45°) (N = 50 trials). (**C**) Facial camera with pupil size detected using DeepLabCut (**left** panel). Average local field potential of the hippocampal silicon probe around the saccade movement of the eyes (**right** panel). The spectrogram shows the spectral average of three channels (each located in stratum pyramidale, stratum radiatum and stratum lacunosum moleculare) around the saccade movement of the eye. (**D**) Cross correlation between each cortical pixel and medium gamma power in the stratum lacunosum moleculate (s.l.m.).

**Figure 4 ijms-23-06814-f004:**
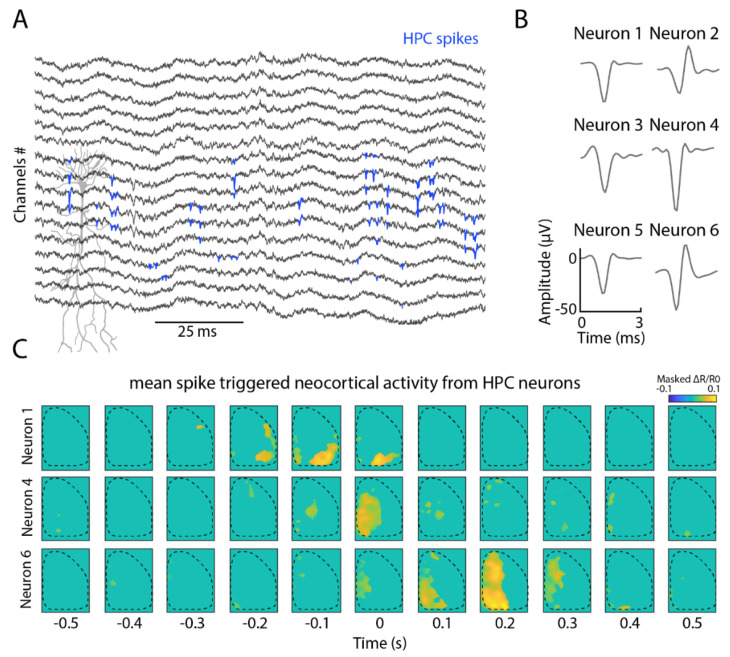
Combined cortical voltage imaging and hippocampal electrophysiological recordings reveal different activity pattern types. (**A**) Local field potential from the silicon probe in the hippocampal CA1. In blue, putative spikes detected using kilosort 2.0. (**B**) Waveforms from 6 example neurons detected in the CA1 pyramidal layer. (**C**) Neocortical activation around spike activity from 3 neurons in the hippocampus. The plots are masked by the mean of the signal plus 5 standard deviations (N = 200 averaged shuffle spikes for each detected neuron).

**Figure 5 ijms-23-06814-f005:**
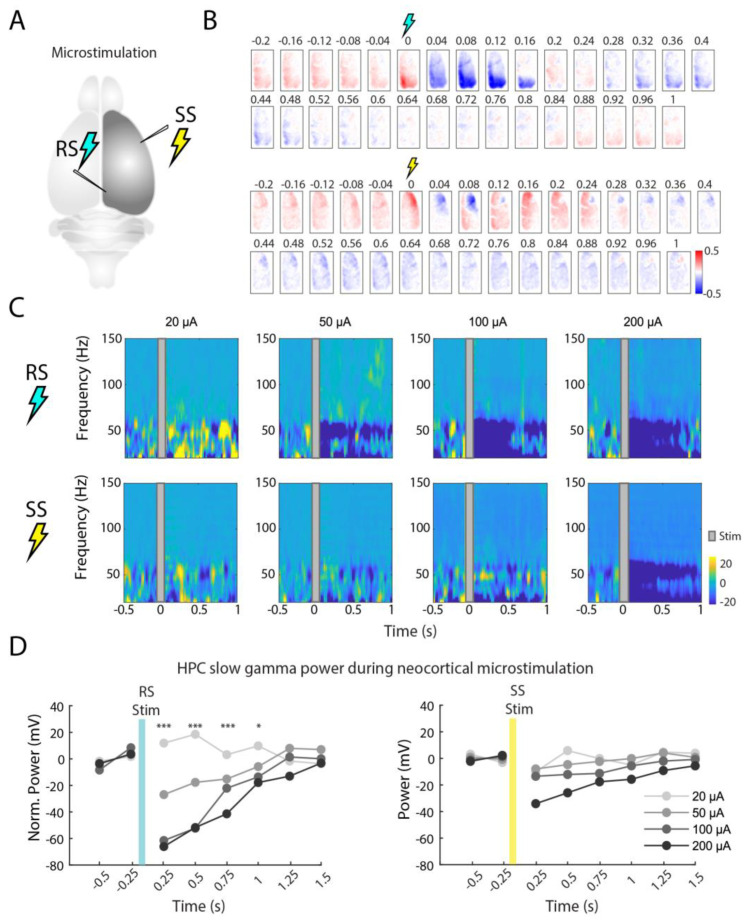
The influence of cortical microstimulation on cortical and hippocampal activities. (**A**) Schematic of the microstimulation zone in the retrosplenial (RS) and somatosensory (SS) cortices. (**B**) Cortical voltage activity patterns following RS and SS stimulations (10 ms stimulation; 200 µA; average of 50 stimulation events). Note that after the short period of increased activity, a continuous suppression takes place during the subsequent second. (**C**) Hippocampal CA1 spectrogram at 20, 50, 100 and 200 µA for the RS and SS stimulation. (**D**) Baseline normalized power of gamma (20–50 Hz) around the 20, 50, 100 and 200 µA stimulations (* *p* < 0.05, and *** *p* < 0.0005, repeated-measures ANOVA, N = 50 stimuli for each condition).

## Data Availability

Data from this study is available from the corresponding author upon reasonable request.

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
