# Peer review of "Combining Cortical Voltage Imaging and Hippocampal Electrophysiology for Investigating Global, Multi-Timescale Activity Interactions in the Brain"

_ijms, 2022, doi:10.3390/ijms23126814_

Round 1

Reviewer 1 Report

The manuscript by Rafael Pedrosa, Chenchen Song, Thomas Knopfel, and Francesco Battaglia (ijms-1778576) describes a new experimental paradigm. Combining a silicone probe and cortical wide-field GEVI recording opens a new way to tackle the questions in brain science. The authors are experts in the field, and the data presented in the manuscript is neat and solid. 

The manuscript should be published in the International Journal of Molecular Sciences.

Please add the appropriate color scale for Figures 2A, 3B, and 3C.

Author Response

We thank the reviewer for the points. In the updated version we added the color scale for the referred figures.

Reviewer 2 Report

In their article, " Combining cortical voltage imaging and hippocampal electrophysiology for investigating global, multi-time scale activity interactions in the brain ", Rafael Pedrosa et al. used functional GEVI to monitor in vivo neural activity with a wide field of view and higher time resolution, and showed collateral evidence of compatibility of neural activity across broad regions by integrating technologies. These experimental results will generate important implications for understanding memory retention and transfer mechanisms between the hippocampus and cortex. And it should be widely recognized work that makes these advanced techniques available to a large number of researchers. Further comments would be added as slight modifications may refine this paper.

Minor issues/comments

1: Last sentence in the introduction, authors are highlighting how this method can be useful for different possible investigations, using the characterization of hippocampal-neocortical interactions as a case study. It should be more clearly stated which parts of the author’s technological development are likely to be particularly new reach, even there are references, [3,13-18].

2: Although it is well understood that the time-resolved analysis using GEVI this time can detect more physiological properties, it is necessary to explain how far the integration of conventional wide-field imaging using GECI and electrophysiology has been shown to date, mentioning references.

3: In the figure1, even if it has been illustrated in the past, it would be easier to understand if the structure and mechanism of this GEVI were described.

4: In the result 2.1, authors mention “coexisting with hippocampal theta, cor-69 tical activity, mostly primary motor cortex (M1), seem more tightly related to running”. It might be more interesting and recognizable here if multiple results were integrated and indexed rather than just a parallel of representative waveforms.

5: Similar to the above, in the result 2.2, in addition to the time series spike heatmaps (Figure 4c) it might be helpful to make some score indicating the compatibility of the gamma-band activity in the stratum lacunosum moleculare, with local circuits in the hippocampus.

6: Even though it has been illustrated in the past, it would be easier to understand if the structure and mechanism of the GEVI in this study were described. Also, any advantages of the chiVSFP over others should be mentioned.

7: It would be better to mention again in the conclusion what this analysis has revealed and add future prospects.

Author Response

Reviewers are in normal font and our reply is in bold font

In their article, " Combining cortical voltage imaging and hippocampal electrophysiology for investigating global, multi-time scale activity interactions in the brain ", Rafael Pedrosa et al. used functional GEVI to monitor in vivo neural activity with a wide field of view and higher time resolution, and showed collateral evidence of compatibility of neural activity across broad regions by integrating technologies. These experimental results will generate important implications for understanding memory retention and transfer mechanisms between the hippocampus and cortex. And it should be widely recognized work that makes these advanced techniques available to a large number of researchers. Further comments would be added as slight modifications may refine this paper.

Minor issues/comments

1: Last sentence in the introduction, authors are highlighting how this method can be useful for different possible investigations, using the characterization of hippocampal-neocortical interactions as a case study. It should be more clearly stated which parts of the author’s technological development are likely to be particularly new reach, even there are references, [3,13-18].

We have revised the introduction clarify our intention to show how the combination between wide-field voltage imaging with hippocampal electrophysiology can contribute to studying cortical-hippocampal interaction specifically during behavior.

2: Although it is well understood that the time-resolved analysis using GEVI this time can detect more physiological properties, it is necessary to explain how far the integration of conventional wide-field imaging using GECI and electrophysiology has been shown to date, mentioning references.

We thank the reviewer for this point. We have added the following into the introduction to clarify this point:

“In particular, conventional wide-field imaging using GECI presents a significant temporal discrepancy in activity when compared with electrophysiology, due to the distinct nature of the recorded signal (REF1; REF2)”

REF1: https://www.nature.com/articles/s42003-021-01670-9

REF2: https://journals.plos.org/ploscompbiol/article?id=10.1371/journal.pcbi.1008198

3: In the figure1, even if it has been illustrated in the past, it would be easier to understand if the structure and mechanism of this GEVI were described.

We added in figure 1 a schematic of the breeding used for the GEVI line of mice we are using (CaMK2A-tTA;tetO-chiVSFP). We amended the text by a more detailed description of the structure and mechanism of the GEVI used and referenced a schematic depiction. See also response to point 6.

4: In the result 2.1, authors mention “coexisting with hippocampal theta, cor-69 tical activity, mostly primary motor cortex (M1), seem more tightly related to running”. It might be more interesting and recognizable here if multiple results were integrated and indexed rather than just a parallel of representative waveforms.

The reviewer is correct. We added in the figure the calculation of the cross correlation between theta power in the hippocampus with the cortical activity in each individual pixel.

5: Similar to the above, in the result 2.2, in addition to the time series spike heatmaps (Figure 4c) it might be helpful to make some score indicating the compatibility of the gamma-band activity in the stratum lacunosum moleculare, with local circuits in the hippocampus.

For this, we calculated the cross correlation between each pixel in the cortex with the medium gamma in the stratum lacunosum moleculare. Additionally, we also comment on the finding in the discussion and results sections.

6: Even though it has been illustrated in the past, it would be easier to understand if the structure and mechanism of the GEVI in this study were described. Also, any advantages of the chiVSFP over others should be mentioned.

Please also see our response to point 3. We amended the text with the text:

 “chiVSFP was developed from earlier VSFPs by replacing a segment of the Ciona intestinalis voltage sensing domain by a homologous portion of the fast activating and deactivating Kv3.1 potassium channel. This modification accelerated the GEVI response dynamics, with chiVSFP readily following membrane voltage oscillations at frequencies up to at least 200 Hz. In contrast to monochromatic GEVIs, chiVSFP reports membrane voltage changes in two fluorescence bands with opposite fluorescence changes. This allows for ratiometric measurements that facilitate correction for hemodynamic and pH related confounds inherent to monochromatic (single wavelength) GEVI imaging.”

7: It would be better to mention again in the conclusion what this analysis has revealed and add future prospects.

We added a paragraph at the end of discussion to conclude, focusing on future prospects.